

# Effect of simulated clinical use and sterilization on the cyclic fatigue resistance of nickel titanium files

Mohammad Alajemi and Ammar AbuMostafa

College of Dentistry, Department of Restorative Dentistry, Riyadh Elm University, Riyadh, Saudi Arabia

## ABSTRACT

**Aim**. Assess the effect of simulated clinical use and sterilization on the cyclic fatigue resistance of Race Evo and Tia Tornado Blue nickel titanium (NiTi) files.

**Materials and Methods**. For this study, a total of sixty-four NiTi files were selected, with thirty-two files each from two different manufacturers. Files from each manufacturer were subdivided into four subgroups ($n = 8$) based on the test parameters. The control groups included files that were neither used nor sterilized. Files from the test groups were used to prepare the root canals of extracted mandibular premolars and then sterilized. This procedure was repeated once, twice, or thrice, depending on the test group. All files were then subjected to a cyclic fatigue test. Data was statistically analyzed using the Kruskal–Wallis and Mann–Whitney U tests.

**Results**. No significant difference was observed in the number of cycles to failure (NCF) among the subgroups for both types of files ($P = 0.869$ for Tia Tornado Blue, $P = 0.626$ for Race Evo). Tia Tornado Blue files displayed significantly higher NCF values in the control ($P = 0.021$), once ($P = 0.027$), and thrice ($P = 0.031$) usage groups when compared to Race Evo files.

**Conclusions**. Repeated clinical use and sterilization for up to three cycles did not affect the cyclic fatigue resistance of Race Evo and Tia Tornado Blue files.

## INTRODUCTION

Due to their improved flexibility, efficiency, and cutting capabilities, nickel titanium (NiTi) rotary instruments have become popular for root canal preparation. The mechanical properties of these files have been improved through thermal treatment methods using gas atmospheres or salt baths (*Pereira et al., 2015*).

While navigating a root canal, NiTi instruments may break due to cyclic fatigue caused by tensile forces on the outer curvature, and compressive forces on the inner curvature. Torsional fatigue, which occurs when the file rotates while its tip remains immobilized, may also contribute to file separation. These situations may occur independently or concurrently (*Peters, 2004*).

Microcracks emerge on the surface of the file due to cyclic fatigue pressures, and typically start as minor flaws on the instrument's outer surface. These microcracks may lead to file separation inside the canal without any visible signs of distortion. Fatigue failure may also

Corresponding author
Ammar AbuMostafa,
ammarabumostafa@riyadh.edu.sa

result from geometrical discontinuities, porosities, inclusions, and overheating of the files during production (*Yilmaz et al., 2018*).

It is hypothesized that the fracture of NiTi files largely happens through a single overload incident during preparation, rather than as the result of significant alloy fatigue after repeated loading cycles. To overcome this drawback, thermomechanical treatments are performed to produce files with enhanced fatigue resistance, flexibility, and centering ability (*Alapati et al., 2004*).

Although the manufacturer's general advice is to only use these instruments once, they are often sterilized and reused.

Many researchers have studied the impact of sterilization on the mechanical characteristics of NiTi files; however, the outcomes were mixed (*Hilfer et al., 2011*; *Plotino et al., 2012*; *Sharroufna & Mashyakhy, 2020*; *Kim et al., 2020*). Some scholars have shown that cyclic fatigue resistance increases as a result of sterilization (*Silvaggio & Hicks (1997)*; *Mize et al., 1998*), while others have encountered conflicting findings (*Viana et al., 2006*; *Plotino et al., 2012*).

Race Evo (FKG Dentaire, Switzerland), a sequenced NiTi file system, was developed in 2020 for performing minimally invasive root canal preparations. The files are manufactured from heat-treated NiTi alloy and electropolished. Likewise, the other tested system, Tia Tornado Blue NiTi files (TiaDent, Houston, TX, USA), are heat-treated firewire rotary files. Thus, evaluating these systems seems to be appropriate for assessing their clinical benefit. These two types of NiTi files were selected for this study due to their similarities in metallurgy, dimension, and triangular cross section.

To date, several published studies have evaluated the impact of autoclaving on cyclic fatigue resistance. However, our literature search revealed that there are no published articles evaluating the effect of both sterilization and clinical use on the cyclic fatigue resistance of Race Evo and Tia Tornado Blue NiTi files. Hence, we aimed to test these parameters.

The null hypothesis was that there would be no significant differences in cyclic fatigue resistance among the groups within each NiTi file system after clinical use and sterilization.

## MATERIALS AND METHODS

### Ethical approval

The research was registered at the Research Centre of Riyadh Elm University and received approval from the Institutional Review Board (IRB) (No. FPGRP/ 2021/595/532/516). Consent forms were not necessary as this was a laboratory-based study with no human subjects. Additionally, the teeth used in this study had been previously extracted for unrelated orthodontic or periodontic purposes. The teeth were collected from the university's dental clinics.

### Sample power calculation

The sample size was calculated using G*power. Differences between two independent means (Race Evo *vs* Tia Tornado Blue) were analyzed using a priori approach to calculate the sample size. Given an alpha of 0.05 and assuming a medium effect size of 0.2, the

calculated sample size was 64. A total of 64 NiTi Files were taken and divided into two groups (Race Evo *vs.* Tia Tornado Blue). These were subdivided into four groups of 8 files each, which was sufficient with a statistical power of 0.84 or 84%, assuming a cutoff point of 0.8 (G*power 3.1.9.7).

## Groups

A total of 64 NiTi files (#25 taper 06 and 25-mm length) were evaluated in this in-vitro study. The files were divided into two groups based on their manufacturers: Race Evo (FKG Dentaire, Sàrl, Switzerland) and Tia Tornado Blue (Tiadent, Houston, TX, USA). Each group ($n = 32$) was further subdivided into 4 subgroups as follows: *Control* ($n = 8$): files neither used nor sterilized. *Once* ($n = 8$): files used and sterilized one time. Each file in this group was used to prepare a root canal for one tooth. *Twice* ($n = 8$): Files were used and sterilized twice. Each file in this group was used to prepare root canals for two teeth. *Thrice* ($n = 8$): files were used and sterilized three times. Each file in this group was used to prepare root canals for three teeth.

## Teeth selection

Extracted human mandibular premolars were chosen according to the following criteria: Type I in Vertucci's classification, curvatures ranging from 20 to 30 according to Schneider, closed apex, and not previously treated nor initiated. Teeth with more than one apical foramen, a calcified canal, resorption, and open apex were excluded.

## Sample inspection

Before the experiment, all instruments were microscopically inspected for morphological deformities or defects using an operational microscope with $\times 25.6$ magnification (Zumax OMS2350; Dental Microscope, China). All examined files were found to be free of defects, and none were discarded. Radiographs were taken of the teeth to confirm they were Type I.

## Simulation of clinical use

The teeth were cleansed with distilled water, disinfected with 1% sodium hypochlorite (NaOCl), and kept in a 0.1% hyaluronate solution. The roof of the pulp chamber along with any overhanging dentine was removed while gaining coronal access using a round size 4 bur (Dentsply-Maillefer, Tulsa, OK, USA). The pulp chamber's roof was then lifted using an Endo Z bur (Dentsply-Maillefer, Tulsa, OK, USA) to refine the cavity. A 2.5% NaOCl solution was used to irrigate the pulp chamber. The exact root canal length was determined using a #10 K-file (Dentsply- Maillefer, Tulsa, OK, USA), which was inserted into the canal until the tip was visible at the apical foramen. This measurement was decreased by one mm and a #15 K-file (Dentsply-Maillefer, Tulsa, OK, USA) was used to determine the final working length. The root canal was then prepared at room temperature using the rotary file according to the manufacturer's recommendations in terms of speed, torque, and technique: 800 rpm at 1 Ncm of torque for Race Evo, and 350 rpm at up to 2 Ncm of torque for Tornado. The file was cleaned with gauze, and the canal was rinsed with 2 ml of 2.5% NaOCl after every three in-and-out strokes inside the canal. A total of six to nine in-and-out strokes were used to reach the working length. A single file was used to prepare

one root canal and then discarded. After each root preparation, the instrument was dried for 15 min, and depending on the categorized group (once, twice, and thrice), sterilized in the autoclave (CISA 420, Italy) at 121 °C at a pressure of 30 psi for 20 min. For the groups of twice and thrice usage, the above procedure was repeated two or three times, respectively. The files were then put through a fatigue test.

## Cyclic fatigue test

All instruments were subjected to continuous clockwise rotational testing at room temperature to determine their susceptibility to cyclic fatigue. A testing apparatus was fabricated for this experiment, consisting of a fatigue-testing stainless steel block attached to the main frame. The steel block contained a simulated root canal that was 25 mm long, with a single curvature of 60°, radius of five mm, and depth of 1.5 mm. The block was fabricated in accordance with the instructions provided by *Larsen et al. (2009)* and *Haïkel et al. (1999)*. A 25 mm long size 15 K-file was inserted to measure the canal's length. A glass plate was placed over the steel block in order to secure it inside the canal, visualize the operation of the file, and enable the removal of shattered pieces.

The main frame was connected to an electric rotating handpiece, which was then precisely positioned in a replicable connection to the steel block's artificial canal. Each file was placed perpendicular to the opening in the centre of the man-made canal. To ensure that the glass cover would not be in contact with the files, they were installed in the device and precisely placed at the same spot in the testing block. Lubrication spray (Pegasus Hand Piece Lubrication Spray) was applied to minimize frictional heat. The rotary files were rotated using a 16:1 reduction handpiece and an electric motor (X-Smart; Dentsply Maillefer, Tulsa, OK, USA) at the respective system manufacturer's specified speed. The cyclic fatigue tests were carried out by a single operator.

The instruments were turned until a fracture was audible or visually evident. In order to prevent human error, a video recording was made concurrently using a mobile phone camera (iPhone 12; Apple, Cupertino, CA, USA), and the recordings were used to confirm the exact moment of file separation. The time to fracture was recorded in seconds. The number of cycles to fracture (NCF) was calculated by the following formula:

$$NCF = \frac{time\ to\ fracture\,(s) \times motor'\ speed\ of\ rotation\,(rpm)}{60}.$$

In addition, an electronic micrometer caliper with an LCD screen was used to measure the length of the files' damaged portions (Inch and Millimeter Conversion, Adoric).

## Scanning electron microscopy

Scanning electron microscopy (SEM) investigation of ten files fractured by cyclic fatigue was conducted to determine the topographic pattern of the fractured files. Five files were randomly selected from the subgroups of both manufacturers (Race Evo and Tia Tornado Blue). Samples were cleaned using 30% ethanol for 6 min in an ultrasonic cleaner to remove any debris. Following cleaning, the samples were vertically mounted using a unique holder. SEM Jeol JSM-6610LV (Jeol, Tokyo, Japan) imaging was performed with 15 kV at 230X magnification to assess the topographical properties.

## Statistical analysis

The Shapiro–Wilk test analysis of the data's distribution revealed a skewed distribution ($p < 0.001$). Calculated descriptive statistics included mean and standard deviations for the duration (seconds) and quantity of cycles used to divide the files into various categories. The Kruskal-Wallis test was used to compare the average number of cycles before fracture, the number of seconds, and the broken section in terms of millimeters between the four groups. The mean cycles to fracture, number of seconds, and broken section in millimeters were compared across the four groups using the Mann–Whitney U test. The statistical analysis was performed using SPSS version 21 (Armonk, NY: IBM Corp., USA). Statistical significance was defined by a $p$-value of 0.05.

## RESULTS

When the subgroups of control, once, twice, and thrice were compared for the same file system, no significant difference was found in the Number of Cycles to Failure (NCF) for Tia Tornado ($P = 0.869$) or Race Evo ($P = 0.626$) (Table 1).

NCF was significantly higher in the subgroups of control ($P = 0.021$), once ($P = 0.027$), and thrice ($P = 0.031$) for Tia Tornado files in comparison to Race Evo files (Table 2).

The mean length of the broken instruments was 5.2 mm ($\pm$ 0.67) without significant differences among the groups.

A scanning electron microscope (SEM) was used to observe distinct indications of cyclic fatigue fracture on the shattered surface. All instruments had fractured surfaces with microvoids and morphologic characteristics of ductile fracture (Fig. 1).

## DISCUSSION

Many practitioners reuse files for financial reasons (*Bird, Chambers & Peters, 2009*). The goal of this study was to assess how sterilization and clinical use affected the cyclic fatigue resistance of Race Evo and Tia Tornado Blue NiTi files.

Studies in the literature have yielded contradictory results regarding the repeated reuse of rotary NiTi files, particularly in terms of their efficiency and physical properties. Some studies reported that these instruments could be safely used up to 10 times in both straight and curved canals (*Peters & Barbakow, 2002*; *Gambarini et al., 2012*). Other studies reported that repeated clinical use of the instruments, which are subjected to sterilization each time, reduced the instruments' physical properties such as cyclic fatigue resistance, cutting efficiency, and flexibility (*Bahia & Lopes Buono, 2005*; *Seago et al., 2015*; *Kamali & Turkaydin, 2021*).

The process of sterilization is necessary for all endodontic files. Several studies were conducted to evaluate the effect of sterilization on different NiTi rotary files. Some studies chose 10 cycles (*Hilfer et al., 2011*; *Plotino et al., 2012*) as a standard to study the effect of sterilization on these files. However, three cycles were chosen for this study, aligning closely with the manufacturer's recommendation to reduce the use of the instruments and minimize the likelihood of file separation.

**Table 1 NCF of subgroups for both file systems.**

| NCF | Control Mean ± SD | Once Mean ± SD | Twice Mean ± SD | Thrice Mean ± SD | P-value[§] |
|---|---|---|---|---|---|
| Tia Tornado | 654.4 ± 128.0 | 613.9 ± 137.8 | 599.7 ± 180.5 | 586.9 ± 117.9 | 0.869 |
| Race Evo | 517.9 ± 94.3 | 509.4 ± 48.1 | 469.6 ± 85.2 | 475.5 ± 60.8 | 0.626 |

Notes.
[§] P-value has been calculated using Kruskal Wallis test.

**Table 2 Comparison between NCF of subgroups in both file systems.**

| Subgroups | NCF Tia Tornado Mean ± SD | NCF Race Evo Mean ± SD | P-value[§] |
|---|---|---|---|
| Control | 654.4 ± 128.0 | 517.9 ± 94.3 | 0.021[*] |
| Once | 613.9 ± 137.8 | 509.4 ± 48.1 | 0.027[*] |
| Twice | 599.7 ± 180.5 | 469.6 ± 85.2 | 0.115 |
| Thrice | 586.9 ± 117.9 | 475.5 ± 60.8 | 0.031[*] |

Notes.
[§] P-value has been calculated using Mann–Whitney U test.
[*] Significant at $p < 0.05$ level.

Mandibular premolars from humans were used in this study. This approach was designed to address the issues raised by the earlier researches in which acrylic blocks were used. To assess the cyclic fatigue resistance of NiTi equipment through simulated clinical use, many earlier studies proposed and used a dynamic cyclic fatigue model. This model has certain constraints. The test equipment was not restricted to follow a specific trajectory. While the pace and amplitude of the axial movements might be standardized in a dynamic system, it is unlikely that these variables will be constant and reproducible in a clinical setting because they are entirely subjective (*Keskin et al., 2017*). As such, the static model was chosen for this study to reduce the confounding factors caused by instrument separation mechanisms other than cyclic fatigue. There was also no need to simulate the clinical situation using a dynamic model because the files utilized in this study were used to create canals on naturally extracted teeth.

No significant differences were observed among the tested subgroups of Tia Tornado Blue and Race Evo files. Thus, the null hypothesis was accepted. When the two file systems were compared, the NCF of the control, once, and thrice usage groups of Tia Tornado were significantly higher than those of Race Evo. Additionally, there were no significant differences in the length of broken fragments among the tested groups.

In this study, Tia Tornado Blue files were found to have better resistance to cyclic fatigue than Race Evo files. Although both file systems are heat–treated, have controlled shape memory, have the same size and taper, and have a triangular cross-section, the heat treatment method and metallurgy could be different, and manufacturers have their own specific methods. However, this finding could be attributed to the high rotational speed of Race Evo (800 rpm) compared to Tia Tornado (350 rpm). A recent study indicated that the resistance of NiTi rotary files to cyclic fatigue is inversely proportional to the rotational speed, it found that files rotating at 200 rpm and 350 rpm are significantly more resistant to

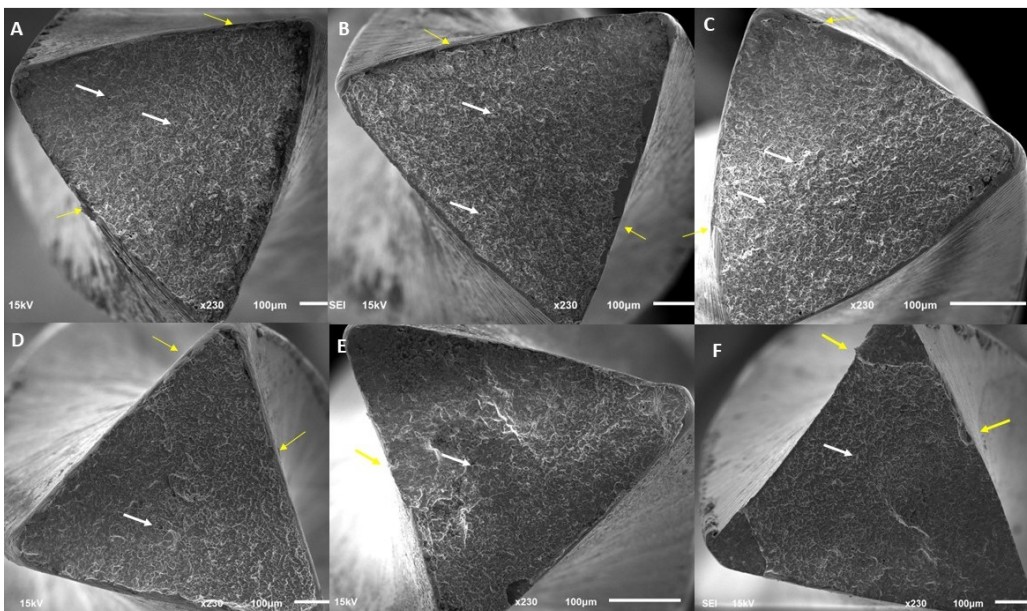

**Figure 1** **SEM images showing the fractured surfaces of files after cyclic fatigue testing.** (A, B, C) Tia Tornado Blue groups of once, twice and thrice, respectively. (D, E, F) Race Evo groups of once, twice, and thrice, respectively. Yellow arrows indicate propagation points at the margins, white arrows indicate dimples on the surface.

cyclic fatigue failure compared to the same files when rotating at 500 rpm (*Faus-Matoses et al., 2022*). Conversely, another recent study testing the cyclic fatigue resistance and bending of the files used in this study (Race Evo and Tia Tornado Blue files) found that Race Evo files are more resistant to cyclic fatigue failure than Tia Tornado Blue files (*Ramadan, AbuMostafa & Alharith, 2023*). This inconsistency between findings could be attributed to the difference in the methodology. The dynamic cyclic fatigue test was used in the study by Ramadan et al., while the static cyclic fatigue test was used in this study.

The clinical use and sterilization of the files in this study had no impact on the tested files' cyclic fatigue resistance. Some studies have claimed that heat-treated NiTi files exhibited improved cyclic fatigue resistance following sterilization (*Plotino et al., 2012*; *Zhao et al., 2016*). This could be explained by the heat treatment applied to the NiTi files and the thermomechanical processing applied to the final product (*Zinelis et al., 2007*). Recent research found that the cyclic fatigue resistance of Race Evo files was unaffected by autoclave sterilization (*Almohareb et al., 2021*). However, after being exposed to sterilization, the heat-treated files exhibit improved characteristics. Furthermore, heat-treated instruments may be exposed to additional thermal treatment during the sterilizing process, increasing their flexibility (*Dioguardi et al., 2020*).

A systematic review and network meta-analysis was performed to assess the influence of sterilization procedures on the physical and mechanical properties of rotating endodontic instruments, an apparent contrast among the results of included studies was found. This could be due to the large variation in the methodology of the studies, such as different

devices used to measure cyclic fatigue, different angles of curvature of the artificial canals, different metallurgy and geometry of the files tested, and different numbers of sterilization cycles. In conclusion, there is a potential shape recovery effect and increased cyclic fatigue resistance after autoclaving. However, not all studies agree (*Dioguardi et al., 2021*)

Another systematic review study evaluated the influence of autoclave sterilization procedures on the cyclic fatigue resistance of heat-treated nickel-titanium instruments, concluding that autoclave sterilization procedures appear to influence the cyclic fatigue resistance of these NiTi files (*Silva et al., 2020*)

During clinical use, root canal instruments are subjected to a variety of stressors. The level of stress concentration and the possibility of instrument fracture can be influenced by both instrument design and instrumentation technique (*Xu et al., 2006*; *Berutti et al., 2012*). Shen et al. reported that 3.4% of instruments were deformed after multiple clinical uses (*Shen et al., 2013*).

Previous investigations have found that cyclic fatigue resistance may decrease following simulated clinical use (*Bahia & Lopes Buono, 2005*; *Pessoa, da Silva & Gavini, 2013*; *Arias, Perez-Higueras & de la Macorra, 2014*). According to an analysis by Duque et al., the cyclic fatigue resistance of instruments produced by ProDesign R and WaveOne Gold (VDM, Munich, Germany) was impacted by simulated clinical use, but this was not observed in instruments produced by Reciproc Blue (*Duque et al., 2020*). Another study found that multiple reuses had no effect on REC25 files' torsional behavior but had a considerable negative impact on their cyclic fatigue resistance when utilized in four or more molar canals (*Pedullà et al., 2022*).

To prevent the tiny variations in file location from affecting the cyclic fatigue behavior, the artificial canal was built with dimensions comparable to those of the tested files. With a mean of 5.2 mm (SD 0.67), the length of fractured fragments was discovered to be near the greatest curvature of the artificial canal, supporting earlier research. This finding demonstrates that the point of maximum stress was comparable in each scenario, verifying that the file was placed with a precise trajectory (*Jamleh et al., 2019*).

When new and used equipment for the two file systems were analyzed using SEM, the typical fractographic characteristic of cyclic fatigue was visible. The instruments displayed crack initiation sites, overload (rapid fracture) zones, and numerous dimples on the cracked surface following the cyclic fatigue test (*Elnaghy & Elsaka, 2016*).

One limitation of this study was that cyclic fatigue resistance was assessed after its use in only one type of canal. Thus, the findings of this study may not be applicable to other canal morphologies. Another limitation of this study was that the experiment was conducted at room temperature, while the phase transition between martensite and austenite in most heat-treated files occurs at a higher temperature. Testing the files below this temperature would not simulate the advantages of using these heat-treated files as they will be mostly or totally in the austenite phase, which might lead to clinically irrelevant results.

## CONCLUSION

Three rounds of simulated clinical use and sterilization did not affect the cyclic fatigue resistance of Tia Tornado and Race Evo NiTi files. However, Tia Tornado files seem to be less affected by clinical use and sterilization compared to Race Evo files. When both systems were compared, the NCF of Tia Tornado files was generally higher than that of Race Evo with statistical significance in some subgroups.

### Funding

The authors received no funding for this work.

### Competing Interests

The authors declare there are no competing interests.

### Author Contributions

- Mohammad Alajemi conceived and designed the experiments, performed the experiments, analyzed the data, prepared figures and/or tables, authored or reviewed drafts of the article, and approved the final draft.
- Ammar AbuMostafa conceived and designed the experiments, analyzed the data, prepared figures and/or tables, authored or reviewed drafts of the article, and approved the final draft.

### Data Availability

The raw data is available in the Supplemental File.

### Supplemental Information

Supplemental information for this article can be found online at http://dx.doi.org/10.7717/peerj.17418#supplemental-information.

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
