# Peer review of "Effect of simulated clinical use and sterilization on the cyclic fatigue resistance of nickel titanium files"

_PeerJ, doi:10.7717/peerj.17418_

## Round 0.1 · original submission · Major Revisions

Your manuscript has been revised and offers some major aspects that have to be revised. Kindly provide a detailed point-by-point answer to reviewers' comments.

**Language Note:** The review process has identified that the English language must be improved. PeerJ can provide language editing services - please contact us at [email protected] for pricing (be sure to provide your manuscript number and title). Alternatively, you should make your own arrangements to improve the language quality and provide details in your response letter. – PeerJ Staff

Reviewer 1 ·

Basic reporting

Good English, background and citation and, all clear

All fine in this category

Experimental design

The research within the scope
The design is clear and been used in many studies and reproducible
Research questions were answered and adding some knowledge the present lit
All fine in this category

Validity of the findings

Valid and valuable results using good stats and clear conclusion
All good in this part

Additional comments

Generally, good work by the authors, I just do have some comments

1. In the abstract: (Files from the test groups were used
and sterilized once, twice, and thrice, respectively, to prepare root canals of extracted human mandibular premolars.) NOT VERY CLEAR THE WAY YOU PUT IT …. MAKE IT CLEARED
2. LINE 140 ( was repeated another one or two times accordingly) REPHRASE
3. Line 242 (It is known that the higher the rotational speed, the less resistance to cyclic fatigue fracture (16). THE REF WAS TALKING ABOUT TORQUE NOT SPEED, FIND THE CORRECT REF OR REMOVE THE SENTENCE or reassure the information
4. Line 246 (fatigue resistance following sterilization (7,21). REF 21 IS NOT RELATED
5. LINE 234 (the null hypothesis was rejected) Did not mention the null hypothesis in the Inro and in the discussion reported that the null hypothesis was rejected? ADD TO THE INTRO
6. You can add a paragraph in the discussion illustrating that the different and conflict results of different files ,despite all heat treated and similar or slightly different methodology been used IS EXPECTED because the heat treated method and metallurgy , design , alloy are quite different and manufactures have their own secrets.

Reviewer 2 ·

Basic reporting

I would like to thank the authors for their study, it follows the correct structure criteria with original findings within the scope of the journal.
However certain points and issues need to be addressed.
The manuscript in general and the Discussion section in particular suffers from weak and repetitive phrases. English language editing and proof-reading is required.

Title: Perhaps the title could be more specific.
Abstract:
1) Abbreviations in the abstract are not explained: Niti and NCF.
2) The term “Brand” while true has a commercial tone. It denotes that the purpose of the study was to compare two commercial brands. Suggest use a different term.
3) Data analysis and significance missing in abstract.
Introduction:
1) The meaning of “Niti” abbreviation is not explained in the introduction.
2) The reason for choosing the two specific files in this study was not explained and why comparing them is of interest.
3) What is meant “without any sign” in line 67.
4) A description of the files and their metallurgy should be provided in the introduction and how they compare to other files on the market.

Experimental design

Materials and Methods:
1) The sample size calculation paragraph should be re-written. The power is “estimated” at 84%. What about the α- and the effect size?
2) Line 106: repetition: plz consider to rephrase: “root canal for their individual tooth”
3) Mentioning number of teeth within the groups description is a bit confusing. Plz consider defining the number of teeth if necessary, in separate sentences.
4) Was blinding of the operator who carried out the cyclic fatigue testing considered? How would it be achieved when the groups were named with the names of the files and the number of sterilization and uses?!!!
5) Why was Schneider technique used to determine curvature? It is well known that the radius and position of the curvature is as important as the degree. This would have affected the standardization of the canals used in the study.
6) The sample in this study are the files and not the teeth. The title “sample preparation” detailing how the teeth were prepared is confusing. The preparation of the canals with the files should be under the title: Simulation of clinical use.
7) It is not clear from the material and methods the sequence of niti file use: was 25/.06 taken directly to the full WL after the k-file 15? How was instrument use inside the canal standardized? Were the number of strokes limited/counted or the time which the instrument remained in the canal measured/limited?
8) Was preparation done at room temperature?
9) What were the sterilization parameters: temperature -cycles?
10) The cyclic fatigue test was run at room temperature?
11) What was the interest of choosing three magnifications so close to each other (170X, 200X and 230X) The images in the figure show only the 230X.

Validity of the findings

Images:
1) The SEM images show only two cross-sections of files: which files are they (after how many uses and sterilization)? Why were not all groups shown and compared?
2) The images should also contain arrows or demarcations for the characteristics of ductile failure in the file sections: crack initiation site, dimpling etc.
3) Why wasn’t a longitudinal SEM image of the file shown?

Tables:
1) Non-parametric tests were used in this study according to the material and methods and not the anova and t-tests.

Raw data:
There were certain errors in calculating the NCF in the raw data: Tornado blue specimen 1 in the control group - Race evo specimen 8 in the control group. While these two errors may not influence statistical significance, Re-calculation of the means is necessary.

Results and Discussion:
1) Do the manufacturers of race evo and tornado blue recommend single use? Plz check with manufacturer instructions.
2) Although the experiment simulated certain aspects of clinical use, it failed to simulate use at a temperature close to that of body temperature 37⸰C. This is of major importance and has significantly influenced your results. For race evo and many heat-treated files, the phase transition between martensite and austenite occurs at a temperature of 35-37⸰C. Testing the files below this temperature would not simulate the advantages of using these heat-treated files and lead to clinically irrelevant results.
3) How do the authors explain that the mean NCF of Race evo in the control group was significantly lower than that reported in a previous study, kindly refer to: https://doi.org/10.3390/met11121947
4) The statement on line 242- 243 while true, applies to conventional niti files and not heat treated files. Please refer to studies on heat-treated files.
5) No need for the word “hot” line 249.
6) What is meant by “more modern” in line 250.
7) What was the interest of comparing the number of uses 1,2 and 3. A five or ten uses group would have served as a positive control.
8) Length of fracture fragment was not addressed in the discussion.
9) Please rephrase the conclusion to avoid “seem not to”.

Reviewer 3 ·

Basic reporting

I would like to thank the authors for their submission. The article is regarding an interesting topic on the relationship between the effect of clinical use and the sterilization process on cyclic fatigue resistance. Despite the interesting topic, the manuscript needs some major revisions reported below.
English language editing and proof-reading is required.

Abstract:
• Kindly specify the abbreviation
• Kindly specify data and statistical analysis in the abstract

Introduction:
The introduction clearly describes the background of the research, however, the reason for instruments selection should be described in more detail.
Kindly add a null hypothesis in the introduction and modify the discussion accordingly.

Experimental design

• Sample size calculation is not clear. How did the Authors determine it (A priori, from literature, etc.)? Effect size 84%?
• Why did the authors use the Schneider?s method and not the Pruett one to determine the canal curvature? It has been demonstrated that also the radius of curvature has a key role in cyclic fatigue.
• Kindly specify in more detail the instrumentation protocol for each group, it is not clear.
• Kindly specify if the cyclic fatigue test was performed at intracanal temperature or at the ambient one. Maintaining the temperature at 35°C seems to be crucial for cyclic fatigue test. Kindly discuss this point.
• Kindly specify the settings of the sterilization procedures.

Validity of the findings

Results:
• Kindly provide one SEM image for each group also containing the description of the image

Kindly improve the comparison of your findings with the current literature.

---

## Round 0.2 · Major Revisions

Considering the reviewers' comments, I kindly ask the authors to review their paper according to them. A point-by-point response is strongly recommended. Major revision is required.

Reviewer 1 ·

Basic reporting

Good English, background and citation and, all clear

Experimental design

The research within the scope
The design is clear and been used in many studies and reproducible
Research questions were answered and adding some knowledge the present lit
All fine in this category

Validity of the findings

Valid and valuable results using good stats and clear conclusion
All good in this part

Additional comments

AUTHORS ADDRESSED MY COMMENTS

Reviewer 2 ·

Basic reporting

Major English editing is required before the manuscript can undergo further evaluation.

Experimental design

Can the authors please explain their choice of an effect size of 0.84? What is the estimated power of the study?
Please redo and upload the tables according to the new statistical analysis done.
Were the fracture segments lengths not normally distributed? please re-check.

Validity of the findings

Discussion still falls short of clear explanation of results.

---

## Round 0.3 · Major Revisions

Please address the remaining issues of the reviewer and amend the manuscript accordingly.

Reviewer 2 ·

Basic reporting

I would like to thank the authors for their work on the manuscript which is now both clear and well-presented.
Most of the points have been addressed, but there are still some that need further attention:

Nickel and not nickle
-please write the formula for calculating the NCF in a clear way.

Experimental design

-How were the ten files selected for SEM? please elaborate.

Please indicate the Shapiro wilks test result on line 174.

Why were the tests conducted for the time? The speed is different between the two files.

Line: 107: Please correct the KV setting for the SEM to 15 (as indicated in the images)

line: 236 ” A recent study indicated that the resistance of NiTi rotary files to cyclic fatigue is inversely proportional to the rotational speed” please re-examine the article as the study was comparing reciprocating and continuous rotation files had a different conclusion.

Validity of the findings

Line: 236 ” A recent study indicated that the resistance of NiTi rotary files to cyclic fatigue is inversely proportional to the rotational speed” please re-examine the article as the study was comparing reciprocating and continuous rotation files with a different conclusion.

A recent study: https://doi.org/10.3390/coatings14010015, in addition to that of Ramadan et al (cited in your paper) have both reported a mean NCF for Tia tornado blue far lower than the one you reported. How can you explain that?
-Please discuss the fact that the testing was not conducted under simulated-body temperature and how that affects your findings.

---

## Round 0.4 · accepted · Accept

All remaining concerns of the reviewer were addressed and revised manuscript is acceptable now.

Reviewer 2 ·

Basic reporting

The authors have satisfied all the comments. Thank you.

Experimental design

The authors have satisfied all the comments. Thank you.

Validity of the findings

The authors have satisfied all the comments. Thank you.